# Suicide and Suicide Prevention Activities Following the Great East Japan Earthquake 2011: A Literature Review

**DOI:** 10.3390/ijerph191710906

**Published:** 2022-09-01

**Authors:** Masatsugu Orui

**Affiliations:** 1Sendai City Mental Health and Welfare Center, Sendai 980-0845, Japan; oruima@fmu.ac.jp; Tel.: +81-24-547-1180; 2Department of Public Health, Fukushima Medical University School of Medicine, Fukushima 960-1295, Japan

**Keywords:** Great East Japan Earthquake, Fukushima Daiichi nuclear power plant accident, suicide, suicide prevention, literature review

## Abstract

Background: Since the Great East Japan Earthquake (GEJE), numerous studies have been conducted, but no comprehensive review study has been carried out. Thus, this literature review aimed to examine how the GEJE might affect suicide and suicidal behaviors from a long-term perspective. Methods: For the literature review, a search of electronic databases was carried out to find articles written in English and in Japanese that were related to suicide and its risk factors, as well as suicide prevention activities following the GEJE. Thirty-two articles were then selected for the review. Results: There were several findings, as follows: (1) gender differences in suicide rates in the affected area: nationwide, the suicide rates in men showed a delayed increase, whereas suicide rates in women increased temporarily immediately after the GEJE; (2) the suicide rates increased again in the recovery phase; (3) the background of the suicides was linked to both disaster-related experiences, and indirect reasons pertaining to the GEJE; and (4) intensive intervention combined with a high-risk and community-focused approach could prevent suicides following the disaster. Conclusions: Although further accumulation of knowledge about suicide and suicide prevention is essential, these findings can contribute to response, recovery, and preparedness in relation to future disasters.

## 1. Introduction

The Great East Japan Earthquake (GEJE), which occurred on 11 March 2011, was the largest in Japanese history. The earthquake generated a massive tsunami that reached a maximum height of 9.3 m, travelling up to 10 km inland across flat areas and causing severe damage to the affected areas in the Iwate, Miyagi, and Fukushima prefectures. This was followed by a separate tsunami, which hit the Fukushima Daiichi Nuclear Power Plant operated by the Tokyo Electric Power Company, causing radiation disasters in the Fukushima Prefecture. Thus, the disaster caused many to lose family members, experience severely damaged homes, and face harsh conditions after evacuation. The experience has affected the psychological and mental health of residents and, as a consequence, could lead to suicidal behavior.

However, according to a previous systematic review study, natural disasters have been shown to have differing effects on suicide rates (e.g., a slight increase or decrease immediately after the disaster, a delayed increase, and no association with the disaster) [1]. Therefore, a consistent understanding has not yet been reached regarding changes in suicide rates after a devastating disaster. Furthermore, the backgrounds of those who died by suicide or who attempted suicide after devastating disasters are still uncertain, although mental disorders such as depression and post-traumatic stress disorder (PTSD), low social support, and disaster-related experiences such as injury to the person and their relatives were identified as the most important risk factors for suicide ideation based on previous cross-sectional studies [2]. Other review articles have reported that natural disasters increase social ties in affected communities, which may mitigate some of the adverse consequences and result in a decline in suicide rates [3]. It has also been reported that prevention strategies which focus on lower socio-economic strata have the potential to have similar effects as those which target psychiatric risk factors [4].

Since the GEJE, numerous studies have been carried out, i.e., descriptive studies of suicide rates and those who attempted suicide, cross-sectional studies of suicide ideation, and disaster-related mental health activities related to suicide prevention. However, no comprehensive review study has yet been conducted regarding suicide and suicide prevention activities following the GEJE. Thus, this literature review aims to examine, from a long-term perspective, how the GEJE might affect suicide and suicide-related behavior in the affected areas. Our goal was to gather together the findings of these studies and thereby contribute to the response, recovery, and preparedness for future disasters.

## 2. Materials and Methods

### 2.1. Definition of the Disaster: The Great East Japan Earthquake

The GEJE occurred at 14:46 JST on 11 March 2011, triggering massive tsunami waves that travelled up to 10 km inland and caused enormous damage to the Pacific Coast in the Iwate, Miyagi, and Fukushima prefectures. This was followed by a separate tsunami, which hit the Fukushima Daiichi Nuclear Power Plant operated by the Tokyo Electric Power Company, causing nuclear disasters in Fukushima Prefecture. It was not only the impact of the disaster itself, but also the dramatic changes in living conditions caused by the evacuation, that had a psychological impact on the residents. Therefore, in this literature review, the definition of the GEJE includes: (1) the massive earthquake and tsunami disaster, and the first large-scale nuclear disaster in Japan; and (2) the aftermath caused by the earthquake, tsunami, and nuclear disasters.

### 2.2. Study Process for the Literature Review

#### 2.2.1. Database

For this literature review, we used PubMed and “Ichushi-Web”, which is a database with an exhaustive collection of Japanese biomedical literature. Specifically, we searched for articles written in English or Japanese which were related to suicide and its risk factors, as well as suicide prevention activities following the GEJE, and which were published between March 2011 and December 2021.

#### 2.2.2. Search Strategy and Inclusion/Exclusion Criteria

The search for English articles was conducted on 1 January 2022, using PubMed, and this was implemented using the following keywords: (1) “the Great East Japan Earthquake” and “suicide”; and (2) “Fukushima nuclear” and “suicide”. Japanese articles were searched via the “Ichushi-Web” using the following keywords: (1) “Higashi Nihon Daishinsai (the Great East Japan Earthquake)” and “Jisatsu (suicide)”; and (2) “Fukushima houshasen (nuclear, radiation)” and “Jisatsu”.

Original articles, short reports and communications, letters, and practical or activity reports that included new findings regarding suicide or suicidal behaviors met the criteria for selection. Commentary and special feature articles were excluded from the literature review because they did not contain sufficient new findings regarding suicide or suicidal ideation or behavior following the GEJE. Duplicate articles listed in both the PubMed and Ichushi-Web databases, and the same articles searched by different keywords (“the GEJE” or “Fukushima Nuclear”), were also excluded.

#### 2.2.3. Literature Selection Process

The initial search, using the proper combination of keywords (“the Great East Japan Earthquake” and “suicide”/“Fukushima” and “suicide” in PubMed and “Higashi Nihon Daishinsai” and “Jisatsu”/“Fukushima houshasen” and “Jisatsu” in Ichushi-Web), led to the identification of 180 articles (41 in English and 139 in Japanese). Subsequently, after screening the identified articles, 17 English articles were excluded because they were commentary, special feature or duplicated articles for the two keywords “GEJE” and “Fukushima”. Similarly, 128 Japanese articles were excluded because they were almost all commentary and special feature articles or duplicated by those listed in PubMed. Next, 35 articles (24 in English and 11 in Japanese) were examined in terms of their content; 3 articles were excluded because they included no direct outcomes related to the GEJE and the Fukushima Daiichi Nuclear Power Plant accident (i.e., the main outcome did not relate to the GEJE but to the Kumamoto Earthquake of 2016; the monitoring of long-term suicide rates ran from 2001 to 2014 and did not include the effect of the GEJE; and the research methods were unclear). Finally, 32 articles (21 articles in English and 11 articles in Japanese) were selected for this literature review (Figure 1).

#### 2.2.4. Categorizing Selected Articles

The selected articles that met the inclusion/exclusion criteria were categorized as follows: (1) descriptive epidemiological study about regional suicide rates in the affected area or other areas; (2) ecological study examining the factors related to the suicide rates; (3) cross-sectional or longitudinal study examining the factors related to suicidal ideation or behavior; (4) descriptive study concerning those who attempted suicide and visited emergency hospitals; (5) case reports; (6) practical reports regarding suicide prevention activities and disaster-related mental health activities; and (7) review study of counselors’ daily records and training interventions for the counselors.

### 2.3. Ethical Considerations

Not applicable as this was a literature review study.

## 3. Results

After the selection and categorization of the articles, seven categories were identified concerning the GEJE and suicide and suicide prevention activities (Table 1). In this results section, selected articles from each category are summarized and integrated.

### 3.1. Descriptive Epidemiological Study about Regional Suicide Rates in the Affected Area or Other Areas

In total, 13 articles were categorized as descriptive epidemiological studies, of which 4 targeted the changes in suicide rates in non-affected areas, nationwide and in Tochigi Prefecture, which is located in the south of Fukushima.

According to the nationwide studies, suicide rates in women increased for several months immediately after the GEJE [5,6] whereas, conversely, the rates in men showed a large decline for several months following the GEJE [5,6], and suicide by men aged 40–64 years and 65 years and over exhibited a large decline in the years following the GEJE, a decline which attenuated over time [7]. In Tochigi Prefecture, located in the south of Fukushima Prefecture, suicide rates in 2011 increased and peaked in May, which contrasted with the years 2008, 2009, and 2010, which saw rates peak in March [8].

In three severely affected areas in the Iwate, Miyagi, and Fukushima prefectures, articles that analyzed suicide rate trends for a period of 2–5 years after the GEJE reported that the suicide rate of men in the disaster-affected areas initially decreased significantly following the GEJE [9,10,11,12], but then showed a delayed increase [10,11,12]. Although one study reported that the female suicide rate declined slightly initially, and then increased significantly over the subsequent 3-year period [12], it was not clear whether there was an association between the GEJE and suicide rates in the affected area comparable to that for males. The overall suicide rate decreased initially, then increased from 2–3 years after the GEJE [11,13].

For a period of over 5 years following the GEJE, known as “the recovery phase”, a couple of articles reported that the suicide rate in men increased again, which was probably because of the termination of financial aid for evacuees in need and the cutting of social ties amongst evacuees [13,14,15].

With regard to disaster-related suicide, which was defined as individuals who died by suicide who were either evacuees or directly affected by the GEJE (as reported by the Cabinet Office), men accounted for a large proportion of suicides in the year in which the disaster occurred, but the proportion of suicides among women increased two years after the disaster [16]. The main motives for suicides were health problems, which increased significantly after the GEJE [16]. The 99 disaster-related suicide cases defined by the Cabinet Office in Fukushima were notably higher in men than in women, whereas disaster-related suicides among women in their 50s and 80s were much higher compared with overall suicide rates in either Fukushima or Japan [17].

### 3.2. Ecological Study Examining the Factors Related to the Suicide Rates

According to two studies which examined the association between the suicide rates in the affected area and related factors, the suicide rates in the affected prefectures, namely Iwate, Miyagi, and Fukushima, did not exhibit a significant association with bankruptcy cases, or the ratio of effective job offers two years after the GEJE [18]. However, the high suicide rate in Iwate Prefecture was found to be related to the small number of temporary houses and small disaster recovery budgets [19].

### 3.3. Cross-Sectional or Longitudinal Study Examining Factors Related to Suicidal Ideation or Behaviors

In three cross-sectional studies regarding suicidal ideation among young girls and adults, both groups showed a significant association with disaster-related experiences such as loss of their home, completely or partially damaged homes, and loss of family or relatives. These studies were conducted in the three-year period following the GEJE [20,21,22]. However, one study that targeted junior high school students showed no significant association with disaster-related experiences, and this was conducted five years after the disaster, indicating that the psychological effects of the disaster may have subsided over time [23].

Other factors related to suicidal ideation were identified as risks related to the following: not being married, poor subjective physical health, suffering from a mental illness, and onset of chronic or delayed mental illness among adult evacuees [21,22].

### 3.4. Descriptive Study Concerning Those Who Attempted Suicide and Visited Emergency Hospitals

According to the two descriptive studies concerning attempted suicides, the number of suicide attempts during the serious conditions of the post-disaster period were significantly higher than in the pre-disaster period, in both the affected and non-affected areas, and this continued for several months [24,25]. In another article, the clinical features of the suicide attempt cases during a four-week period were characterized as acute stress, depressive state, and secondary stress reaction, for example, caused by drastic environmental and lifestyle changes [26]. However, another article reported that the background psychosocial factors related to suicide attempts were variable (e.g., family problems, intersexual problems, and alcohol), and had little direct connection to the GEJE, even though the individuals were all evacuees [27].

### 3.5. Case Reports

One case report article reported that “Since a few months after the disaster, many people have perceived that they are going to lose their neighborhood support networks and have diminished hope for the future without any concrete financial support provided by the government”. This was based on a study which involved searching newspaper articles [28].

### 3.6. Practical Reports Regarding Suicide Prevention Activities and Disaster-Related Mental Health Activities

We identified four practical reports regarding disaster-related mental health and suicide prevention activities, which were implemented in coastal municipalities in Miyagi Prefecture and a municipality in the evacuation area as a consequence of the nuclear disaster in Fukushima.

In Higashi-Matsushima City, Ishinomaki City, and Kesennuma City in Miyagi Prefecture, a year after the GEJE, those among the almost 1500 evacuees who were in a depressive state, e.g., PTSD, anxiety disorders, insomnia, alcoholism, or suicidal ideation, were referred to psychiatrists or public health nurses in the public health centers [29].

In Iitate Village, which was within the evacuation area as a result of the Fukushima Daiichi Nuclear Power Plant accident, some suicide prevention measures were implemented as follows: (1) screening of evacuees for mental health issues and outreach to those who were at high risk; (2) gatekeeper training for public office workers; and (3) supervision and support for public health nurses. Although the direct cause was unknown, the number of suicides in Iitate did not exhibit a sharp increase after the GEJE, with the number staying flat at 0–2 per year pre- and post-disaster [30].

In Sendai City, Miyagi Prefecture, a disaster mental health team was established following the GEJE, and it provided psychosocial and social support to address the psychosocial issues of evacuees on an ongoing basis. Although the direct effect of the disaster on mental health was unknown, at the time of the report the suicide rate in Sendai had not increased since the GEJE [31]. Furthermore, during the COVID-19 pandemic, the suicide rate in the affected area in Sendai, which was provided with disaster-related mental health interventions, showed a continuous declining trend. Conversely, there were increasing trends in suicide rates nationally and in the non-affected area of Sendai [32].

### 3.7. Review Study of Counselors’ Daily Records and Training Interventions for Counselors

According to two articles, the percentage of counselors who were involved in helping residents who had suicidal ideation increased considerably, from 21.1% one year after the GEJE to more than 50% after two years [33,34]. As a result, seminars and gatekeeper training were conducted to help counselors cope with residents’ mental health issues and suicidal problems and as a result, their confidence to cope with the residents’ suicidal ideation showed a significant increase [35,36].

**Table 1 ijerph-19-10906-t001:** Articles selected for the literature review of the GEJE and suicide and suicide prevention activities.

Authors	Measures of Suicidal Behavior	Time Period	Location	Subjects	Analysis	Main Findings
(1). Descriptive epidemiological study about regional suicide rates in the affected area/other areas (13 articles)						
Liu Y. et al. (2015) [5]	Suicide rates (annual and monthly)	A year after the GEJE; comparison of 2005 with 2011	Nationwide	All residents in Japan	Prais–Winsten regression using 11 dummy variables (January taken as the reference) to detect any seasonal variations separately for men and women.	There was no significant difference in any 2 years adjacent to 2011 for either gender. However, with respect to women, the suicide number in 2011 was higher than that in 2010; in particular, the 10 months from March to December 2011 showed a significant difference compared with 2010 for women but not for men.
Osaki Y. et al. (2021) [6]	Suicide rates (SMR, monthly)	2 years in 2011 and 2020 (at the time of the GEJE and during the COVID-19 pandemic in 2020)	Nationwide	All residents of the prefectures	The 95% confidence intervals for the ratio were calculated to determine any statistically significant increases or decreases in the suicide rate nationwide and in the three affected prefectures.	After the earthquake, the male suicide rate in March 2011 was 18% lower than the average mortality rate for the previous 3 years. However, it increased by 18% in May and 8% in June, and increased mortality was observed in women.
Matsubayashi T. et al. (2021) [7]	Suicide rates (annual)	8 years after the GEJE and baseline (from 2002 to 2019)	Nationwide	All residents of the prefectures	Event-study estimates of the GEJE on sex–age specific suicide rates.	Using prefecture-level data, suicide by men aged 40–64 years and 65 years and over showed a significant decline in the years following the GEJE, and this decline attenuated over time.
Tsuboi S. et al. (2012) (in Japanese) [8]	Suicide number and suicide rates (monthly)	From January to June in 2008, 2009, 2010, and 2011	Tochigi Prefecture (next to southern Fukushima)	All residents in Tochigi Prefecture	Time series analysis was used to examine the trend in monthly suicide rates during the same period (January to June) in 2008, 2009, 2010, and 2011.	The monthly suicide rates in Tochigi Prefecture increased in March in 2008, 2009, and 2010; however, it increased from March to May in 2011, peaking in May.
Masaki N. et al. (2018) (in Japanese) [9]	Suicide rates (SMR, monthly, annual)	2 years after the GEJE and baseline (from 2010 to 2013)	The municipalities in the coastal area of the Iwate, Miyagi, and Fukushima prefectures	All residents of the municipalities in the coastal areas of the Iwate, Miyagi, and Fukushima prefectures	The SMRs of the subject area in the affected area were calculated, and then compared with the pre-disaster SMR.	The ratio of the SMR was 0.92 in the first post-disaster period (from 2011 to 2012), and 0.93 in the second period (from 2012 to 2013), which was significantly lower compared with the pre-disaster SMR.
Orui M. et al. (2015) [10]	Suicide rates (monthly, annual)	3 years after the GEJE and baseline (from March 2009 to February 2014)	16 municipalities in the coastal area of Miyagi Prefecture	All residents of the 16 municipalities in the coastal area of Miyagi	Suicide rates were assessed using a time series, in which suicide rates were compared with corresponding national averages.	In tsunami disaster-stricken areas, male suicide rates were initially significantly lower than the national average but began to increase after 2 years. Similarly, male suicide rates in the inland areas decreased for 7 months, and then increased to exceed the national average.
Ohto et al. (2015) [11]	Suicide rates (SMR, annual)	3 years after the GEJE and baseline (from 2010 to 2014)	Disaster affected prefectures (Iwate, Miyagi, Fukushima)	All residents in the affected prefectures	SMR comparison with national average.	SMR decreased during the first 2 years after the disaster in each affected prefecture compared with 2010 (table), and then rose in 2014 to pre-disaster levels in the Iwate and Miyagi prefectures and exceeded the pre-disaster level in Fukushima Prefecture.
Orui M. et al. (2018) [12]	Suicide rates (annual and monthly)	Almost 5 years after the GEJE and baseline (from March 2009 to February 2011)	Fukushima Prefecture, including the evacuation area related to the Fukushima Daiichi Nuclear Power Plant accident	All residents in the Fukushima Prefecture	The exponential smoothing time series model was used to examine the trend of monthly suicide rates. Additionally, period analysis was performed for each 12-month period from March 2009 to February 2015 (and the 9 months from March 2015 to December 2015) and compared to the national average.	Male suicide rates in the evacuation areas increased significantly immediately after the disaster, and then began to increase again 4 years after the disaster. Female suicide rates declined slightly during the first year and then increased significantly over the subsequent 3-year period.
Orui M. (2020) [14]	Suicide rates (annual)	7 years after the GEJE and baseline (from 2009 to 2018)	14 municipalities in the coastal area of Miyagi Prefecture	All residents of the 14 municipalities in the coastal area of Miyagi	Period analysis was used to divide the total 108-month study period into nine segments, in which suicide rates were compared with corresponding national averages using a Poisson distribution.	Male suicide rates in the affected area from March 2013 to February 2014 increased to a level higher than the national average. After subsequently dropping, the male suicide rates increased again from March 2016 to February 2018 and showed a significant difference compared with the national averages.
Orui M. et al. (2020) (in Japanese) [13]	Suicide rates (monthly)	8 years after the GEJE and baseline (from 2009 to 2019)	14 municipalities in the coastal area of Miyagi Prefecture	All residents of the 14 municipalities in the coastal area of Miyagi	Moving average time series analysis was used to remove the seasonal changes.	Male suicide rates increased gradually from May 2016 when the provision of free temporary housing began to be phased out. A little later, from December 2017, female suicide rates also increased.
Kuroda Y. et al. (2021) [15]	Suicide rates (monthly, annual)	8 years after the GEJE and baseline (from June 2009 to December 2018)	Evacuation area in Fukushima Prefecture following the Fukushima Daiichi Nuclear Power Plant accident	All residents of the municipalities in the evacuation area	Time series model using exponential smoothing between the evacuation and the non-evacuation areas in Fukushima Prefecture.	In the evacuation areas, the male suicide rate increased immediately after the disaster and then decreased steeply around 1.5 years after the disaster. However, with the lifting of the evacuation order, it again exceeded that of non-evacuation areas and continued to do so for the next 3 years. On the other hand, the suicide rate in women in the evacuation areas started to increase later than that in men.
Inoue K. et al. (2015) [16]	Disaster-related suicide and its motives	2 years after the GEJE (from 2011 to 2013)	Disaster-stricken prefectures and the prefectures to which evacuees were relocated	All residents of disaster-stricken prefectures and prefectures to which evacuees were relocated	Descriptive analysis of suicide number by gender reported from suicide statistics.	Among the cases of disaster-related suicide, men accounted for a large proportion of suicides in the year that the disaster occurred, but the proportion of suicides among women increased 2 years after the disaster. The main motives for the related suicides were health problems, which increased significantly after the GEJE.
Takebayashi Y. et al. (2020) [17]	Disaster-related suicide numbers and rates	7 years after the GEJE	Fukushima Prefecture	99 suicide cases which were determined to be disaster-related suicides	Descriptive analysis of suicide rates by age, gender, means, and occupation among disaster-related suicide cases as defined by the Cabinet Office.	Age-standardized disaster-related suicide rates were notably higher in men than in women. In addition, disaster-related suicide rates in Fukushima were higher in women in their 50s and 80s compared with overall suicide rates in Fukushima or Japan.
(2). Ecological study (2 articles)						
Orui M. et al. (2014) [18]	Suicide rates (monthly)	2 years after the GEJE and baseline (from March 2009 to February 2014)	Disaster-stricken prefectures and neighboring prefectures (Aomori, Akita, Yamagata)	All residents of disaster-stricken and neighboring prefectures	Using the multiple regression model, the association between suicide rates and economic variables was evaluated based on the number of bankruptcy cases and ratio of effective job offers and comparing the disaster-stricken and neighboring prefectures.	In disaster-stricken areas, male suicide rates decreased during the 24 months following the earthquake. Multiple regression analysis showed that bankruptcy cases and ratio of effective job offers were only significantly associated with male post-disaster suicide rates in the neighboring prefectures rather than in the disaster-stricken prefectures.
Shiga Y. et al. (2016) (in Japanese) [19]	Suicide rates (SMR)	Two periods before and after the GEJE (2008–2010 and 2011–2013)	Iwate Prefecture	All residents in Iwate Prefecture	The suicide SMR was calculated for the nine medical health areas in Iwate Prefecture and the correlation between SMR and economic status, medical care, and disaster damage was examined.	The high suicide rate in Iwate Prefecture was found to be related to the small number of temporary houses and small disaster recovery budgets.
(3). Cross-sectional or longitudinal study examining the factors related to suicidal ideation or behaviors (4 articles)						
Fujiwara T. et al. (2017) [20]	Suicide risk (suicidal ideation, self-injury behavior and suicide attempts in a lifetime); MINI-KID	Baseline: 2 years after the GEJE (from September 2012 to June 2013) Follow-up: 3 years later (from July 2013 to May 2014)	The municipalities in the coastal area in the Iwate, Miyagi, and Fukushima prefectures (Control: Mie Prefecture located in the western region of Japan)	198 children aged 5–8 in the Iwate, Miyagi, and Fukushima prefectures (Control: 82 children aged 5–8 in the Mie Prefecture)	A logistic regression analysis between suicide risks and earthquake-related events, exposure to other trauma before the GEJE, psychological distress (K6), and PTSD symptoms (IES-R).	Four or more trauma experiences related to the GEJE was associated with suicide risk only among girls (Odds ratio: 5.74, 95%CI: 0.83–39.6, *p* = 0.076) compared to no trauma experience related to the GEJE. This showed that young girls who experienced earthquake-related trauma at preschool age had a higher suicidal ideation 3 years after the GEJE.
Xu Q. et al. (2018) [21]	Suicidal ideation (WHO-CIDI) version 3.0	3 years after the GEJE (Iwate, Miyagi: June and August 2014, Fukushima: from October 2013 to February 2014, Control: from August to November 2014)	Severely damaged areas in Iwate, Miyagi, and Fukushima (Control: Eastern Japan excluding Kanto region)	More than 2000 adults aged 20 years and older residing in temporary housing in Iwate, Miyagi, and Fukushima (Control: 1850 individuals)	(1) The cumulative incidence of suicidal ideation using the Cox proportional hazard model. (2) A multiple logistic regression analysis was conducted to examine risk factors for the onset of suicidal ideation.	(1) Amongst 1019 respondents, the cumulative incidence of suicidal ideation 1, 2, and 3 years after the earthquake was 1.4%, 2.4%, and 2.8%, respectively—significantly higher than that in the control. (2) Not being married, being injured in the disaster, and poor subjective physical health were associated with suicide ideation.
Morishima R. et al. (2019) [22]	Suicidal ideation (investigator-designed query)	1, 2 and 3 years after the GEJE (May and June in 2012, 2013, and 2014)	Higashi-Matsushima City in Miyagi Prefecture, where serious damage was inflicted by the tsunami	11,855 residents who were 19 years or older, and had enrolled for the national health insurance	A logistic regression analysis was conducted to evaluate the effect of suicidal ideation risk on the onset of mental illness (K6 > 13 points) for three years, adjusting for age, sex, house damage, presence of cohabitants, residence situation, working status, and seeking counselling for mental health.	Many residents, who showed a high risk of suicidality, still suffer from a mental illness, requiring housing, occupation, and psychological support. Chronic or delayed onset mental illness showed a higher risk of suicidal ideation (Odds ratio: 23.3 and 60.6, respectively).
Kawahara K. et al. (2020) [23]	Suicide risk (suicidal ideation, self-injury behavior and suicide attempts in a lifetime); MINI-KID	5 years after the GEJE	Ishinomaki City, Miyagi Prefecture, which was damaged severely by the tsunami disaster	262 students in two junior high schools	A logistic regression analysis between suicide risks and psychological symptoms (PTSD, depression, anxiety), disaster experiences (housing damage or evacuation), and current residents (temporary or own housing).	Disaster experience was not associated with psychological symptoms (PTSD, depression, anxiety) or suicide risk in junior high school students 5 years after the GEJE. The suicide risk appeared to be the same as that in the general population in Japan.
(4). Descriptive study concerning those who attempted suicide and visited emergency hospitals (4 articles)						
Kato K. et al. (2014) [24]	Suicide attempts involving visits to hospital (cases of minor self-injury excluded)	Pre-disaster (from September 2010 to February 2011) and post-disaster (from April 2011 to September 2011)	Kanagawa Prefecture, next to the Tokyo metropolitan area; this was not the disaster area, but was significantly indirectly affected	286 attempted suicides in the pre-disaster period, 306 in the post-disaster period	The psychiatric characteristics of suicide attempts were assessed, with one or two trained psychiatrists making a diagnosis according to the Diagnostic and Statistical Manual of Mental Disorders, 4th Edition DSM-IV).	With regard to mood, anxiety, schizophrenia, and substance-related disorders, no significant difference in the number of patients with psychiatric disorders was observed between the pre- and post-disaster periods. The number of patients in a serious condition in the post-disaster period was significantly higher than that in the pre-disaster period.
Aoki Y. et al. (2014) [25]	The number of suicide attempts at a tertiary medical center which accepts the most critically ill patients	A year after the GEJE and control period (11 March 2010 to 10 March 2012)	The medical center located in the middle-southern area of the Fukushima Prefecture and 58 km west of Fukushima Daiichi Nuclear Power Plant	Individuals who attempted suicide who were transferred to this medical center	The clinical records of all patients who visited the medical center near the nuclear plant from 1 year before to 1 year after the disaster were reviewed (*n* = 981).	The risk of nonfatal suicide attempts (those who survived) using high mortality methods (methods other than self-poisoning and wrist-slitting) was significantly higher, by three to four times, for 4 months after the series of disasters, and then decreased. There was no significant increase in nonfatal suicide attempts using low-mortality methods (self-poisoning and wrist slitting) after the disaster.
Yoshioka Y. et al. (2015) (in Japanese) [26]	Suicide attempts (those who were admitted to a tertiary medical center)	During the 1-year period after the GEJE	Morioka City, Iwate Prefecture, which is a central city in Iwate Prefecture	10 suicide attempt cases who had problems related to the GEJE	A review of clinical records relating to 10 cases with regards to age, sex, methods of suicide, psychiatric diagnosis, and background of suicide behavior.	The clinical features of cases within 4 weeks were classed as acute stress, and after 4 weeks they were classed as secondary stress reactions, such as drastic environmental changes in lifestyle. Depressive states were also noted.
Ikemoto K. et al. (2011) (in Japanese) [27]	Suicide attempts (those who were admitted to the department of psychiatry at a tertiary medical center)	During the 3-month period after the GEJE	Iwaki City, Fukushima Prefecture, close to Fukushima Daiichi Nuclear Power Plant	11 suicide attempt cases admitted to the psychiatry department	A review of clinical records relating to 11 cases with regards to age, sex, methods of suicide, psychiatric diagnosis, and background of suicidal behavior	The background psychosocial factors relating to the suicide attempts varied between cases (e.g., family problems, intersexual problems, and alcohol-related), which almost had little to no direct relation to the GEJE or the nuclear power plant accident, even in evacuees.
(5). Case report (1 article)						
Yamashita J. et al. (2013) [28]	Suicide cases	Within 6 months of the GEJE	Minami-soma City, Fukushima Prefecture	Residents evacuated because of the GEJE and Fukushima Daiichi Nuclear Power Plant accident	Searching the articles in newspapers	In a note that she left behind, a woman wrote that she did not want to bother her son’s family and had chosen to take refuge in the grave. In the months following the disaster, many people perceived that they were going to lose their neighborhood support networks and had diminished hope for the future, in the absence of any concrete financial support provided by the government.
(6). Practical reports regarding suicide prevention activities and disaster-related mental health activities (4 articles)						
Son D. et al. (2015) (in Japanese) [29]	Mental health issues, including suicidal ideation and other social issues	From August 2011 to May 2012	Higashi-Matsushima City, Ishinomaki City and Kesennuma City in Miyagi Prefecture	Almost 1500 evacuees who relocated to temporary housing	Unknown	Evacuees who had experienced a depressive state, PTSD, anxiety disorders, insomnia, alcoholism, and suicidal ideation were referred to a psychiatrist or public health nurse in the public health centers.
Orui M. et al. (2019) (in Japanese) [30]	The number of suicides in the subject area	Almost 6 years after the GEJE	Iitate Village in the evacuation area due to Fukushima Daiichi Nuclear Power Plant accident	All residents in Iitate Village, Fukushima Prefecture, including evacuees who relocated outside the evacuation area	To implement suicide measures, namely: (1) screening for evacuees regarding mental health issues and outreach to those who have been at high risk; (2) gatekeeper training for public office workers; and (3) supervision and support for public health nurses in Iitate.	Although the direct reason was unknown, the number of suicides in Iitate did not exhibit a sharp increase after the GEJE, with the number staying flat at 0–2 per year, pre- and post-disaster.
Orui M. et al. (2017) [31]	Suicide rates	4.5 years after the GEJE	Sendai City in the disaster-affected area in Miyagi Prefecture	All residents in Sendai City	To address the psychosocial issues of evacuees, the disaster mental health team provided psychosocial and social support. The suicide rate in the disaster-affected area was used as one of the outcome indicators of this activity.	Although the direct effect of this disaster-related mental health activity was unknown, at the time of the report the suicide rate in Sendai City had not increased since the GEJE.
Orui M. et al. (2021) [32]	Suicide rates (monthly)	9 years after the GEJE during the COVID-19 pandemic	14 municipalities in the coastal area of the Miyagi Prefecture	All residents of the 14 municipalities in the coastal area of Miyagi	The suicide rate trends were assessed using exponential smoothing time series modelling. The area in question is the disaster-affected area which has seen the implementation of disaster-related mental health activities on an ongoing basis, including during the COVID-19 pandemic.	During the COVID-19 pandemic, the suicide rate in the affected areas showed a declining trend, whereas the national rates and the rates in the non-affected areas showed an increasing trend, although the affected areas were higher than the national average at the beginning of the COVID-19 pandemic.
(7). Review study of counselors’ daily records and training interventions for the counselors. (4 articles)						
Shiragami K. et al. (2013) (in Japanese) [33]	Suicidal ideation	First survey: December 2011 Second survey: May 2012	A municipality in Iwate Prefecture	24 livelihood support counselors	24 counselors were asked whether they had the experience to address evacuees with suicidal ideation.	At the time of the first survey, none of the counselors had the experience to address suicide ideation among evacuees, although 21.1% of counselors had addressed evacuees with suicide ideation at the time of the second survey.
Ueno M. et al. (2016) (in Japanese) [34]	Suicide-related problems (e.g., suicidal ideation)	18-month follow-up of support for evacuees from May 2011	Disaster-affected area in Miyagi Prefecture	1669 evacuees who relocated to temporary housing	A review of records concerning support for evacuees, and evaluation of the safety status, and physical and mental health condition (e.g., alcoholism, depression, suicidal ideation).	More than 50% of those evacuees who had suicidal ideation have continued to experience mental health issues, even after 18 months of support.
Akazawa M. et al. (2016) (in Japanese) [35]	Self-confidence in counselling and responding to residents’ mental health issues	November and December 2014	Two municipalities in the disaster-affected area in Miyagi	101 livelihood support counselors in the disaster-affected area	To evaluate the outcomes of the training seminar, pre- and immediate post-intervention surveys were conducted.	Counselors’ confidence to cope with the suicidal ideation of residents showed a significant increase at the post-intervention survey.
Orui M. et al. (2020) [36]	Self-confidence in counselling and responding to residents’ mental health issues	November and December 2019	Minami-soma City and Iitate Village in Fukushima Prefecture, in the evacuation areas	26 livelihood support counselors from Iitate Village and Minami-Soma City Social Welfare Council participated in the training program	To evaluate the outcomes of the training program, including a pre- and post-survey, and a follow-up survey 2 months after the intervention was conducted.	The program content regarding suicide was as follows: specific ways to communicate with residents who have suicidal thoughts: (a) questioning about the suicidal thoughts, and (b) encouraging a person to seek appropriate professional help. As a result, counselors’ confidence to cope with the resident’s suicidal thoughts showed a significant increase.

## 4. Discussion

### 4.1. Gender Differences in Suicide Rates in the Affected Area and Nationwide in the Period of Two to Three Years

According to descriptive epidemiological studies regarding changes in suicide rates nationwide and in the non-affected prefecture, suicide rates in women increased temporarily immediately after the GEJE [5,6,8], but suicide rates in the affected prefectures and areas, especially in men, showed a decline for almost two years, and then exhibited a delayed increase [9,10,11,12]. This phenomenon of a delayed increase in male suicide rates was a consistent trend in the affected area following the GEJE, and similar trends were observed in previous studies [37,38,39].

The direct reason for the differences between genders in both non-affected and affected areas is unknown, but some studies have suggested that women are more sensitive and susceptible to adverse events [5], and that increases in female suicide rates during social crises is greater and more prolonged [6]. Therefore, the female suicide rates might increase temporarily immediately after the GEJE in the non-affected area. Intensive and long-term mental healthcare services and psychosocial support [6,10], together with the boosting of economic conditions via reconstruction and recovery [9] could help to reduce suicides in the affected area.

That said, previous studies have mainly focused on monitoring suicide rates in the disaster-affected areas, and few studies have examined this while including national data. Our literature review confirmed that the national trend was different from that of the disaster-affected areas. If large-scale disasters occur in the future, it will be important to keep in mind that suicide rates can also increase in non-affected areas as well as in in the disaster-affected areas, in particular among women.

### 4.2. Suicide Rates Increasing Again in the Recovery Phase

Historically, studies monitoring suicide rates have spanned a period of five years at most, but the literature following the GEJE included long-term monitoring studies of seven to eight years [13,14,15]. These studies found that suicide rates, both in the coastal areas severely damaged by the tsunami and in the evacuation area due to the Fukushima Daiichi Nuclear Power Plant accident, increased again during the recovery phase seven to eight years after the GEJE [13,14,15], although the prevalence of post-disaster suicidal ideation showed a decreasing trend according to a previous study [23,40]. In the recovery phase, the provision of temporary housing was terminated, which increased economic hardship for needy evacuees. Additionally, disruption of the social connectedness established in the temporary housing may have had an influence. These findings were similar to those of a previous review study that found building social ties worked positively to prevent suicide and that those from lower socio-economic strata were at risk of suicide [3,4]. These findings represent an important lesson for future disasters and need to be borne in mind even in the recovery phase.

### 4.3. Background to the Case of Suicide and Suicidal Behaviors Following the GEJE

Firstly, according to the case report identified in our literature review, the background of the suicide case was “diminished hope for the future” and “lack of concrete financial support” [28]. Health problems (including mental health issues) as a motive for suicide based on the suicide statistics of the national police agency increased significantly after the GEJE [16]. In the same way as significant changes involving life and work [41,42], the effects of instability, physical problems [42,43,44], living difficulties [31,45], and hopelessness for the future [46] can be major factors in suicides in the immediate to medium-term period following the GEJE. Additionally, suicidal ideation showed a significant association with not only having disaster-related experiences but also the onset of mental illness [20,21,22], which was similar to the findings of the study of suicide motives following the GEJE [16]. Severe cases of suicide attempts increased immediately after the GEJE [24,25], and the background of those who attempted suicide was related to acute stress and secondary stress reactions such as drastic environmental changes in lifestyle [26]. Clearly, acute stress or drastic environmental changes can pose a challenge to the mental health of evacuees. Notably, it was not only the disaster-related experiences that caused suicide attempts but also reasons indirectly related to the disaster [27].

In summary, these findings indicate that acute stress, drastic environmental changes in life, and psychological factors such as hopelessness and the onset of mental illness were thought to be risk factors for suicide following the disaster. However, because of the complex backdrop of suicide, further accumulation of data is considered necessary to examine more fully the causes of suicide and suicidal behaviors after disasters.

### 4.4. Suicide Prevention Measures Following the GEJE

After the GEJE, numerous disaster-related mental health activities were implemented in the disaster-affected areas [29,30,31,32,33,34]. The basic measures were implemented widely in the affected areas based on the official guidelines [47], as follows: (1) high-risk approaches such as screening evacuees with high psychological distress, binge-drinking or other mental health issues, and ongoing counselling involving home visits, and (2) a community approach, e.g., building social ties among evacuees in the community that has been indicated as a protective factor for suicide in a previous review study [3]. Although practical reports regarding disaster-related mental health activities including outcomes of suicide were limited, these interventions for the evacuees in the affected areas might have been effective for suicide prevention [30,31,32]. However, although this kind of intervention was carried out to a certain extent in all disaster areas after the GEJE on an ongoing basis, and without interruption, the suicide rates in all the affected areas in the Iwate, Miyagi, and Fukushima prefectures showed a delayed increase in the medium- to long-term period following the disaster and increased again in the recovery phase. The reason was uncertain; it might have been related to the small disaster recovery budgets that were reported by the ecological study identified in our review [19].

In any case, it was considered necessary to accumulate knowledge for suicide prevention. It is also important to train counselors to help the evacuees with suicidal ideation in order to prevent suicides in the affected areas, because the counselors who worked with the evacuees with suicidal ideation had a relatively high level of experience [35,36].

### 4.5. Limitations and Strengths of This Literature Study

This literature review has some limitations. Firstly, the identified articles were not sufficiently numerous to enable the findings to be summarized the findings, other than in the case of the descriptive epidemiological studies concerning regional suicide rates. In particular, case reports on individuals who died by suicide or who attempted suicide were extremely limited in terms of understanding the background to the suicidal behavior. Secondly, because the search process was implemented by the author, there is a risk of selection bias even though the search strategy and inclusion/exclusion criteria were set out in advance.

Despite these limitations, this literature review has several strengths, one of which is that it included articles written in Japanese. As a result, more articles written in Japanese than in English were selected (e.g., a descriptive study of those who attempted suicide, practical reports regarding suicide prevention and disaster-related mental health activities, and the review study of counselors’ daily records and training interventions for the counselors). It is hoped that more knowledge regarding suicide and suicide prevention can be accumulated for use in any future disaster scenarios.

## 5. Conclusions

As a result of this study which reviewed literature following the GEJE, several findings emerged, as follows. (1) Gender differences in suicide rates in the affected areas: the suicide rates in men initially declined and showed a delayed increase, and this was mirrored nationwide. However, the suicide rate in women increased temporarily in the period of two to three years following the GEJE. (2) Suicide rates in the affected areas increased again during the recovery phase. (3) The background to suicide was associated with disaster-related experiences, including acute stress, secondary stress reactions such as drastic environmental changes in lifestyle, as well as to reasons indirectly linked to the GEJE. (4) Intensive intervention combined with the high-risk and community-focused approach could prevent suicides after the disaster. Although it is essential to accumulate further knowledge about suicide and suicide prevention following disasters, the findings of this literature review can contribute to the response, recovery, and preparedness for future disasters.

## Figures and Tables

**Figure 1 ijerph-19-10906-f001:**
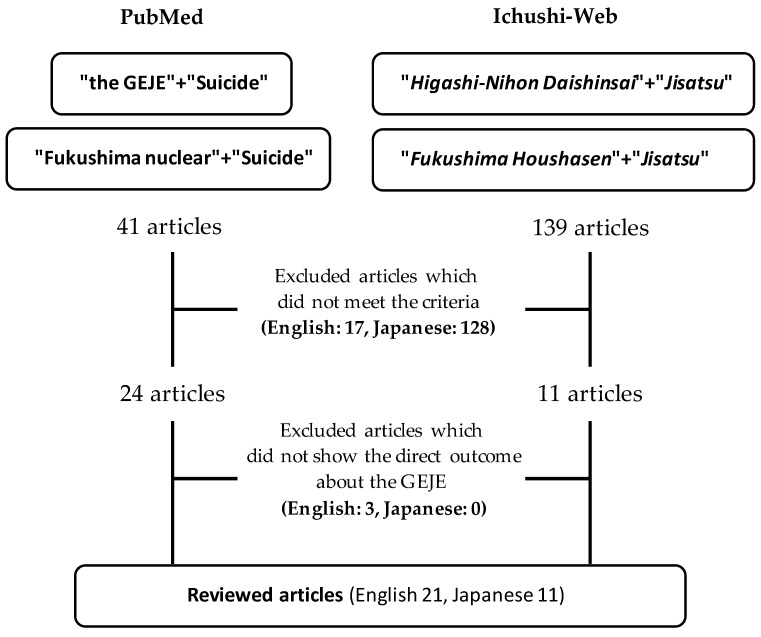
Selection procedure for this literature review.

## Data Availability

Not applicable.

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
