# Peer review of "Suicide and Suicide Prevention Activities Following the Great East Japan Earthquake 2011: A Literature Review"

_ijerph, 2022, doi:10.3390/ijerph191710906_

Round 1

Reviewer 1 Report

The author has done an excellent job to present this review, it provided us with some information about what we should pay attention to at different periods after massive disasters. But there are still some concerns the author should illustrate by adding more literature.

1. According to line 129-130, the suicide rate presented a reverse change mode between men and women, did the overall suicide rate showed any change in the disaster-related area? 

2. Did the suicide rate have an increase after suicide prevention activities or mental health activities discontinued?

Author Response

Thank you for your meaningful comments to help polish my manuscript.
Could you please confirm the revised manuscript and the attached letter?

Reviewer 2 Report

The introduction can be improved.  Review should also examine if there were other studies conducted in other regions, which can support some of the assertions in the discussion. 

Consider the inclusion of some socio-economic data as well.  Somewhat concerned about the approach in that on a temporal scale there could have been subsequent disasters ( though less severe) that could have also contributed. 

Author Response

(The authors gave the same response as above.)
